

# Composition and environmental interpretation of the weed communities in the main planting base of jujube (*Ziziphus jujuba* Mill. cv. 'LingwuChangzao'), Ningxia province of China

Xiaojia Wang[1,2], Bing Cao[1], Jin Zou[1] and Weijun Chen[3]

[1] School of Agriculture, Ningxia University, Yinchuan, Ningxia, China
[2] School of Agriculture, NingXia Polytechnic, Yinchuan, Ningxia, China
[3] Lingwu Natural Resources Bureau, Linwu, Ningxia, China

## ABSTRACT

**Background:** *Ziziphus jujuba* Mill. cv. 'LingwuChangzao' is a traditional jujube cultivar in northwest China. It is of great significance to explore the weed community composition and environmental characterization for the ecological control and comprehensive management of weeds in jujube orchards. In this article, a total of 37 species were recorded in 40 sample plots (1 m × 1 m). Moreover, fourteen environmental indicators to characterize the spatial locations, climate and soil nutrient characteristics of the plant communities were adopted.
**Methodology:** Through the two-way indicator species analysis (TWINSPAN) quantity classification and canonical correspondence analysis (CCA) ranking methods, the types of weed communities in the main planting base of jujube 'LingwuChangzao' and the main environmental factors affecting the change and distribution of weed types were analyzed.
**Results:** The weed communities within the study area were divided into 15 types by the TWINSPAN classification. There were significant differences in soil factors to the species diversity indices of the weed communities, the diversity of weed communities was negatively correlated with available potassium, whereas positively correlated with soil water content. The CCA results showed that community structure and spatial distribution of weed communities were affected by soil water content, total potassium, soil organic carbon, total phosphorus, total nitrogen. Our results can be used as a reference for orchard weed management and provide a theoretical basis for weed invasion control and creating a higher biodiversity in arable land under the background of environmental change.

# INTRODUCTION

The plant community is the product of the interaction between plants and their living environment. Generally, the process is accompanied by the succession of community structure and the change of species diversity (*Wang et al., 2010*). Exploring the ecological

Corresponding author
Bing Cao, bingcao2006@126.com

relationship between plant communities, species and environmental variable will help to reveal the key environmental factors that govern the distribution of species and communities, developing corresponding environmental factor management strategies (*He et al., 2020*).

Weeds are a biological component of agricultural ecosystems. On the one hand, weeds affect the crop yield and quality by competing for light, nutrients, water and living space (*Gandia et al., 2021*). Some weeds are allelopathic and have adverse effects on crops (*Cheng & Cheng, 2015*). Studies have indicated that the root exudates and decomposing substances of *Chenopodium album* and *Capsella bursa-pastoris* induced oxidative damage and cell damage, disturbed the expression levels of key genes related to photosynthesis, thus inhibiting the seed germination and seedling growth of recipient plants (*Li et al., 2018*). On the other hand, weeds are plant resources with potential utilization value (*Tian & Shen, 2012*), and they are also the main component of forest and grass plant diversity that play a key role in maintaining ecological balance (*Sun et al., 2019*). The scientific control of weeds has always been one of the research hotspots of agricultural scholars. Since the second half of the 20th century, chemical herbicides, as one of the important symbols of the green revolution, have been put into farmland (*Li, 2018*). While ensuring the safety of global food production, it has also brought serious problems of weed resistance and farmland environmental pollution (*Li et al., 2019*), and seriously affected the richness of weed species and the genetic diversity of wild plants (*Armengot et al., 2012*). A total of 153 dicots and 113 monocots of herbicide resistant weeds have been found globally, to date (*Heap, 2022*). *Pan et al. (2015)* studied the weed communities with eradicable annual weeds (such as *Echinochloa crusgalli* and *Cyperus difformis*) being predominant have been changing into perennial weed-dominated communities (such as *Alternanthera philoxeroides* and *sagittaria pygmaea*). Therefore, the problem of weeds cannot be solved by blind overuse of herbicides. It is necessary to put forward ecological weed management strategies to keep the weed density within a certain threshold range and maintain a certain weed biodiversity (*Ma et al., 2021c*).

Jujube (*Ziziphus jujuba* Mill.), is one of the oldest fruit trees in China where it has been cultivated for over 3,000 years (*Guo et al., 2021*). This fruit tree is growing in popularity globally for its strong stress resistance, fast fruit, rich nutrition, remarkable economic and ecological benefits (*Wang et al., 2016*), and has been widely cultivated in more than 50 countries in Asia, Europe and America (*Ma et al., 2021b*). The commercial production of *Ziziphus jujuba* Mill. cv. 'LingwuChangzao' is concentrated on the Yellow River Basin in the Ningxia Hui Autonomous Region where jujube cultivation has become an important industry for local farmers to increase income and agricultural efficiency. At the same time, it has also become one of the important means to control soil erosion and improve the ecological environment (*Ma et al., 2021c*; *Zhu et al., 2022*).

Currently, the research on 'LingwuChangzao' jujube mainly focuses on fruit morphological quality (*Jiang et al., 2020*), transcriptome differences analysis of fruit of grafting and root tiller propagation (*Ma et al., 2021b*), irrigation and fertilization patterns (*Tang et al., 2021*), effects of environment factor on sugar accumulation, metabolism and related gene expression in fruit (*Chen et al., 2020*) and storage and preservation (*Yang*
*et al., 2021*). However, the quantitative study on the classification of weeds community, ordination and its relationship with environmental factors in the main production areas of 'LingwuChangzao' jujube has not been reported.

We quantitatively studied the relationship between weed community diversity and environmental factors in the main planting base of 'LingwuChangzao' jujube, by using two-way indicator species analysis (TWINSPAN) and canonical correspondence analysis (CCA). We mainly answer the following questions: (1) classification of clusters and species functional groups of weed community. (2) the effects of heterogeneous environment on species distribution and population characteristics. The final purpose of our research will contribute to understanding the composition, distribution and environmental factors affecting weed distribution in jujube production areas, and provide a theoretical basis for weed invasion control and creating a higher biodiversity in arable land under the background of environmental change.

## MATERIALS AND METHODS

### Study area

This study was conducted at the 'LingwuChangzao' jujube experimental station of the Technical Service Center of Forestry and Fruit Tree, Lingwu City, Ningxia Province, China (37°53′57″–38°13′9″N, 106°19′36″–106°23′32″E, 1110–1180 m in altitude) (Fig. 1). The survey region was of temperate continental climate, with a mean average sunshine of 4434.7 h. Mean annual temperature and rainfall were about 8.8 °C and 206.2–255.2 mm. The mean frost-free period is 157.0 d, the plant growth period lasts 170.0 d, and the effective accumulated temperature is 3351.3 h. The soil types in the study area are mainly sandy loam, irrigation-silting soil, sandy soil and loam. Fruit tree-age ranges from 11 to 18-year old, and the row spacing is dominated by 3 m × 4 m.

### Vegetation sampling

Weed vegetation was investigated in eight typical 'LingwuChangzao' jujube planting bases (S1: Ningxia Yinhu Agrosilvopastoral Technological Development Co., Ltd; S2: Ningxia Xunkunheju Agricultural Development Co., Ltd; S3: Daquan Forest Plantations, Lingwu City, Ningxia Province; S4: Horticultural Farm, Lingwu City, Ningxia Province; S5: Ningxia Daqin Jujube Industry Professional Cooperative; S6: Ningxia Linsenjun Agrosilvopastoral Technological Development Co., Ltd; S7: Ningxia Ningliubao Jujube Industry Professional Cooperative; S8: Shangqiao Village of Linhe Town in Lingwu City) in August 2021. The longitude, latitude and altitude of these plots were measured and recorded by using a handheld GPS system. Five 1 m × 1 m herbaceous quadrats were tossed with a steel quadrat frame within each plot randomly, for a total of 40 samples. The present species, height, abundance, frequency, coverage, biomass of each herbaceous quadrat was recorded.

### Soil sampling and measurement

We collected the soil samples from the top layers (0–20 cm) by utilizing soil auger. We pooled the samples together, generating one mixed sample for each quadrat. Then removed coarse debris, plant roots, stones before laboratory analysis. Soil water content
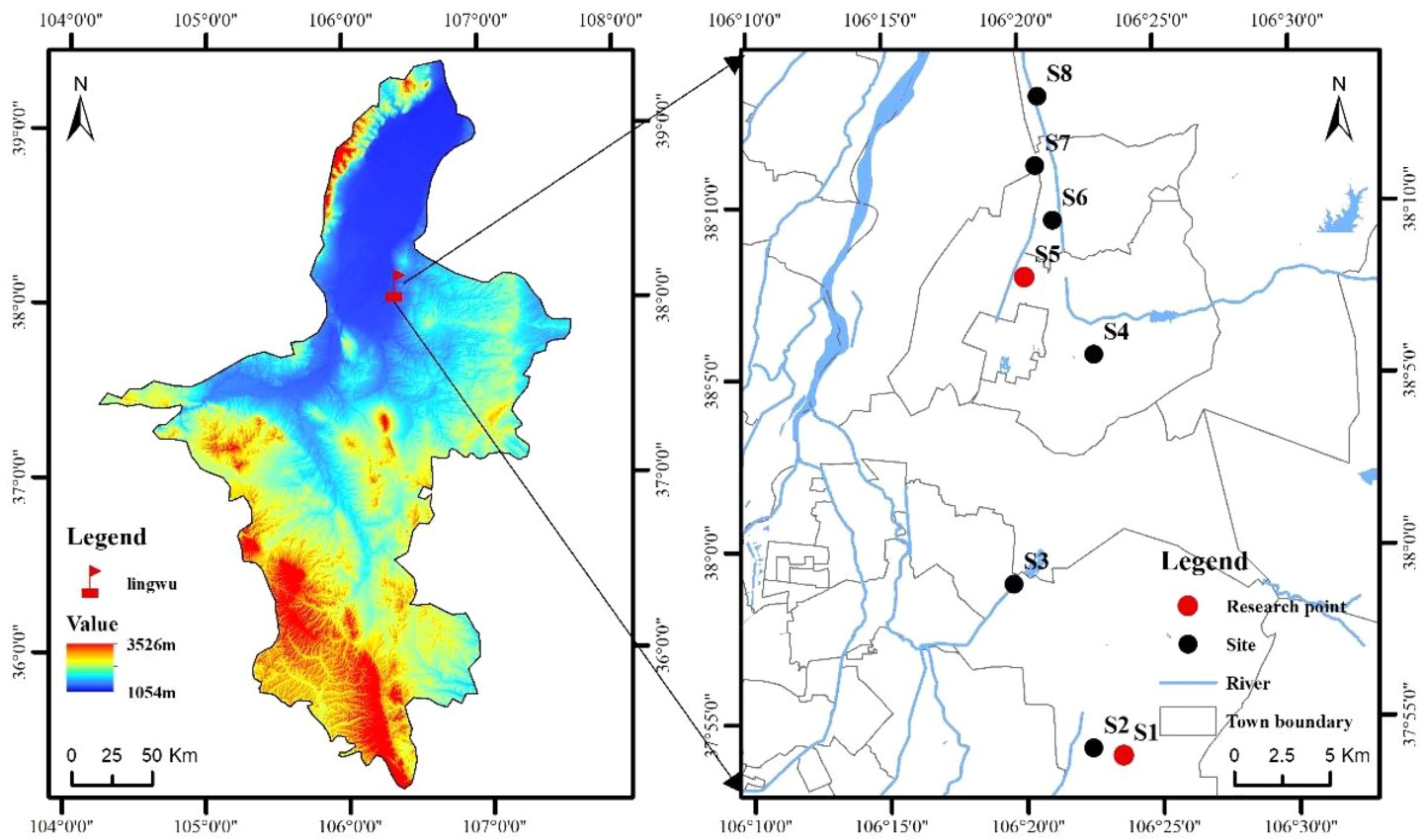

**Figure 1 Sketch map of the research area location and plots setting.**

and soil temperature were measured by temperature and humidity tester (TZS-2X-G). Soil bulk density was determined using cutting ring method. Soil organic carbon (SOC) was determined using the potassium dichromate oxidation method (*Ma et al., 2021a*). Soil total nitrogen (TN) was measured using the kjeldahl method (*Purcell & King, 1996*). Soil total phosphorus (TP) was measured as described by the methods of *Page, Miller & Dennis (1982)*. Soil total potassium (TK) was measured using flame photometry after sodium hydroxide fusion. Soil available phosphorus (AP) and soil available potassium were determined as described by *Wang, Wang & Ma (2022)*. Nitrate ($NO_3^-$-N) and ammonium ($NH_4^+$-N) nitrogen were analyzed by a continuous flow analytical system (AA3: SEAL Company, Röttenbach, Bavaria, Germany) (*Yuan et al., 2021*). The soil pH as well as EC was determined as described by *Bao (2000)*.

## Data analysis

Based on the data of plant community species height, coverage, density, frequency, etc., the species importance value (IV), richness (S), Shannon-Weiner index (H′), Simpson index (D) and Pielou's index (E) were calculated by utilizing following formulas (*Ma, 1994*):

IV $_{herb}$ = (relative coverage + relative height + relative density + relative frequency)/4

Species richness: S = number of species present in the plot

Shannon-Wiener index: $H' = -\sum_{i=1}^{S} P_i \ln P_i$

$$\text{Simpson index: } D = 1 - \sum_{i=1}^{s} P_i{}^2$$

$$\text{Pielou's index: } E = \frac{H'}{\ln S}$$

$$P_i = \frac{N_i}{N}$$

where S is the total number of species in each quadrat; N is the sum of all importance values of S species; $N_i$ is the importance value of species *i*.

The data matrix functions were composed of importance values of 37 species in 40 weed community quadrats (37 × 40), which as the attribute for classification and ranking. WinTwins2.3 was used to divide the weed community species matrix into different association groups. Excel 2016 was used to calculate the diversity indices. The plotting of diversity indices, correlation and multiple comparisons were finished by using Origin 2022. The Canoco5.0 was used to analyze the main soil factors that affect the weed communities and plotted with Cano-Draw. Monte Carlo 499 permutations was used to assess the significance of F-values of environmental variables in CCA.

## RESULTS

### TWINSPAN quantitative classification

A total of 37 species were recorded in 40 quadrats, belonging to 16 families and 36 genera. Compositae (8), Chenopodiaceae (5) and Gramineae (5) account for a relatively high proportion among them. According to the principles of classification and joint nomenclature of plant communities, 40 weed quadrats were divided into 15 association groups by TWINSPAN (Fig. 2). The main characteristics of each weed community in the main planting base of jujube 'LinwuChangzao' are shown in Table 1.

Community I: *Lactucatatarica* (quadrats A13, A14, A29, A32).

This community was mainly distributed in Daquan Forest Plantations, Haojiaqiao Town and Lingwu Farm, Dongta Town, with an average importance value of 0.35 and an altitude of 1,110–1,130 m. The average height of this community was 11.16–14.42 cm, and the total coverage was 23.5–62.3%. The dominant weed was *Lactucatatarica*, and the primary accompanying species were *Cynanchum chinense* and *Chenopodium glaucum*, with average importance values of 0.1 and 0.19 respectively.

Community II: *Portulaca oleracea + Potentilla supina* (quadrats A7, A8, A24).

This community was mainly distributed in Langpiziliang Forest Plantations, Haojiaqiao Town and Dongta Town, with an average importance value of 0.32 and an altitude of 1,120–1,170 m. The average height of this community was 10.59–23.32 cm, and the total coverage was 13–14.1%. The dominant weeds were *Portulaca oleracea* and *Potentilla supina*. The primary accompanying species included *Calystegia hederacea Wall*, *Rumex acetosa*, *Phragmites australis* and *Lepidium apetalum*, with average importance values of 0.18, 0.14, 0.13 and 0.11 respectively.

Community III: *Calystegia hederacea Wall* (quadrats A22, A23, A25).
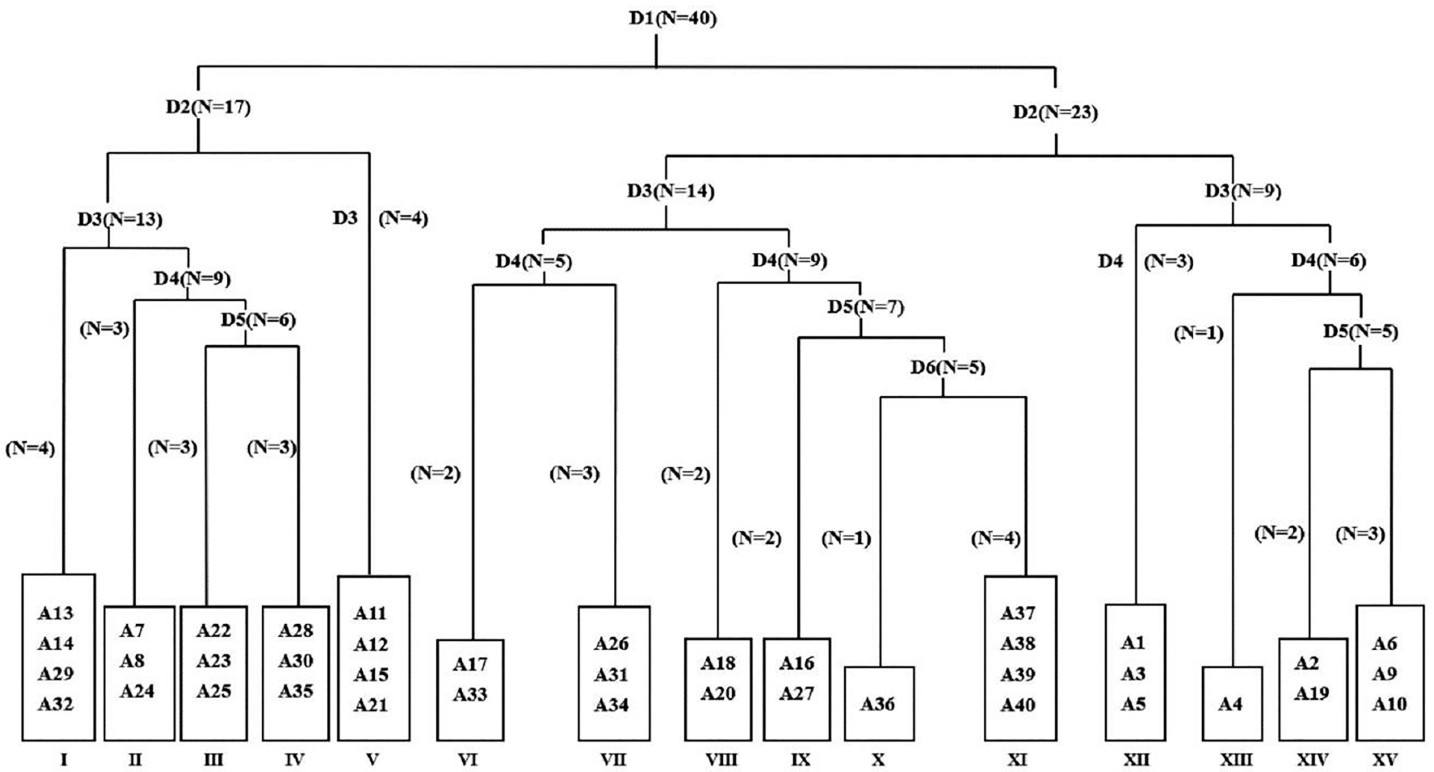

**Figure 2 Taxonomic tree map of 40 plots based on TWINSPAN classification. D represents classification grade; N represents treatment number; A1–A40 represent quadrat codes. I–XV represent community numbers.**

This community was mainly distributed in Dongta Town, with an average importance value of 0.3 and an altitude of 1,120 m. The average height of this community was 25.1–39.2 cm, and the total coverage was 32.1–57.6%. *Calystegia hederacea Wall* was the dominant weed in the community, and the primary accompanying species included *Lepidium apetalum*, *Rumex acetosa* and *Portulaca oleracea*, with average importance values of 0.25, 0.17 and 0.15 respectively.

Community IV: *Lepidium apetalum* (quadrats A28, A30, A35).

This community was mainly distributed in Lingwu Farm, Dongta Town, with an average importance value of 0.34 and an altitude of 1110–1120 m. The average height of this community was 5.54–18.78 cm, and the total coverage was 17–37.5%.
The dominant weed was *Lepidium apetalum*, and the primary accompanying species were *Ixeris chinensis*, *Calystegia hederacea Wall* and *Suaeda glauca Bunge*, with average importance values of 0.26, 0.16 and 0.14 respectively.

Community V: *Rorippa islandica + Chenopodium glaucum* (quadrats A11, A12, A15, A21).

This community was mainly distributed in Daquan Forest Plantations, Haojiaqiao Town and Dongta Town, with an average importance value of 0.31 and an altitude of 1120–1130 m. The average height of this community was 19.3–25.7 cm, and the total coverage was 4.4–73.5%. The dominant weeds were *Rorippa islandica* and *Chenopodium*

**Table 1 Main characteristics of weed communities in the main planting base of jujube 'LingwuChangzao'.**

| No. | Association composition | Altitude (m) | Soil temperature (°C) | Soil water content (%) | Soil bulk density (g cm$^{-3}$) | Important value | Plots |
|---|---|---|---|---|---|---|---|
| I | *Lactucatatarica* | 1110–1130 | 19.8–24.3 | 13.5–27.9 | 1.03 | 0.35 | A13, A14, A29, A32 |
| II | *Portulaca oleracea + Potentilla supina* | 1120–1170 | 18.6–29.5 | 12.4–29.1 | 1.28 | 0.32 | A7, A8, A24 |
| III | *Calystegia hederacea Wall* | 1120 | 19.6–19.8 | 21.1–30.8 | 1.24 | 0.3 | A22, A23, A25 |
| IV | *Lepidium apetalum* | 1110–1120 | 19.5–21.8 | 14.6–29 | 1.1 | 0.34 | A28, A30, A35 |
| V | *Rorippa islandica+ Chenopodium glaucum* | 1120–1130 | 18.6–29.6 | 11.3–29.8 | 1.02 | 0.31 | A11, A12, A15, A21 |
| VI | *Chenopodium glaucum+Ixeris chinensis + Cynanchum chinense* | 1110–1170 | 21.1–31.9 | 9.9–27.9 | 0.98 | 0.84 | A17, A33 |
| VII | *Ixeris chinensis+ Amaranthus retroflexus + Chenopodium glaucum* | 1110–1120 | 19.8–21.2 | 15.6–29 | 1.02 | 0.75 | A26, A31, A34 |
| VIII | *Chenopodium glaucum+ Sophora alopecuroides* | 1170 | 28.6–30.9 | 2.9–8.3 | 1.27 | 0.47 | A18, A20 |
| IX | *Ixeris chinensis+Agropyron cristatum* | 1120–1170 | 19.2–31.9 | 8.1–16 | 1.19 | 0.46 | A16, A27 |
| X | *Ixeris chinensis+ Chenopodium glaucum + Setariaviridis* | 1130 | 20.6 | 14.4 | 1.12 | 0.73 | A36 |
| XI | *Chenopodium glaucum* | 1130 | 20.3–20.6 | 14.1–14.4 | 1.28 | 0.25 | A37, A38, A39, A40 |
| XII | *Digitaria sanguinalis + Chenopodium glaucum* | 1180 | 30.6–38.9 | 7.6–10.4 | 1.21 | 0.74 | A1, A3, A5 |
| XIII | *Digitaria sanguinalis + Chenopodium album* | 1180 | 31.4 | 7.6 | 1.12 | 0.76 | A4 |
| XIV | *Chenopodium glaucum + Digitaria sanguinalis + Ixeris chinensis* | 1170–1180 | 31.4–36.8 | 8.9–10.2 | 1.28 | 0.78 | A2, A19 |
| XV | *Chenopodium glaucum + Digitaria sanguinalis + Portulaca oleracea* | 1170 | 25.8–28.4 | 7.9–21.5 | 1.17 | 0.73 | A6, A9, A10 |

*glaucum*. The primary accompanying species included *Calystegia hederacea Wall* and *Chenopodium album*, with average importance values of 0.14 and 0.13 respectively.

Community VI: *Chenopodium glaucum + Ixeris chinensis + Cynanchum chinense* (quadrats A17, A33).

This community was mainly distributed in Horticultural Farm and Lingwu Farm of Dongta Town, with an average importance value of 0.84 and an altitude of 1,110–1,170 m. The average height of this community was 9.67–34.5 cm, and the total coverage was 12.2–45.7%. The dominant weeds were *Chenopodium glaucum*, *Ixeris chinensis* and *Cynanchum chinense*. The primary accompanying species were *Glycyrrhiza uralensis Fisch*, *Potentilla supina* and *Taraxacum mongolicum*, with average importance values of 0.1, 0.09, 0.09 respectively.

Community VII: *Ixeris chinensis + Amaranthus retroflexus + Chenopodium glaucum* (quadrats A26, A31, A34).

This community was mainly distributed in Lingwu Farm of Dongta Town, with an average importance value of 0.75 and an altitude of 1110–1120 m. The average height of this community was 13.5–24.5 cm, and the total coverage was 19.5–33.8%. The dominant

weeds were *Ixeris chinensis*, *Amaranthus retroflexus* and *Chenopodium glaucum*. The primary accompanying species included *Portulaca oleracea*, *Rumex acetosa* and *Taraxacum mongolicum*, with average importance values of 0.23, 0.13 and 0.1 respectively.

Community VIII: *Chenopodium glaucum + Sophora alopecuroides* (quadrats A18, A20).

This community was mainly distributed in Horticultural Farm of Dongta Town, with an average importance value of 0.47 and an altitude of 1170 m. The average height of this community was 9.28–14.04 cm, and the total coverage was 11.1–38.1%. The dominant weeds were *Chenopodium glaucum* and *Sophora alopecuroides*, and the primary accompanying species included *Setariaviridis*, *Ixeris chinensis*, *Phragmites australis* and *Heteropappus altaicus*, with average importance values of 0.32, 0.26, 0.21 and 0.15 respectively.

Community IX: *Ixeris chinensis + Agropyron cristatum* (quadrats A16, A27).

This community was mainly distributed in Horticultural Farm of Dongta Town, with an average importance value of 0.46 and an altitude of 1,120–1,170 m. The average height of this community was 10.6–12.2 cm, and the total coverage was 16–24%. The dominant weeds were *Ixeris chinensis* and *Agropyron cristatum*, and the primary accompanying species included *Chenopodium glaucum*, *Setariaviridis* and *Incarvillea sinensis*, with average importance values of 0.3, 0.15 and 0.14 respectively.

Community X: *Ixeris chinensis + Chenopodium glaucum + Setariaviridis* (quadrat A36).

This community was mainly distributed in Linhe Town, with an average importance value of 0.73 and an altitude of 1,130 m. The average height of this community was 13.52–26.86 cm, and the total coverage was 22.2–33.3%. The dominant weeds were *Ixeris chinensis*, *Chenopodium glaucum*, and *Setariaviridis*. The primary accompanying species included *Kochia scoparia* and *Portulaca oleracea*, with average importance values of 0.14 and 0.06 respectively.

Community XI: *Chenopodium glaucum* (quadrats A37, A38, A39, A40).

This community was mainly distributed in Linhe Town, with an average importance value of 0.25 and an altitude of 1130 m. The average height of this community was 9.94–19.8 cm, and the total coverage was 15.4–44.4%. The dominant weed was *Chenopodium glaucum*. The primary accompanying species included *Ixeris chinensis*, *Portulaca oleracea*, *Lappula heteracantha*, *Setariaviridis* and *Bassia dasyphylla*, with average importance values of 0.2, 0.17, 0.12, 0.1 and 0.03 respectively.

Community XII: *Digitaria sanguinalis + Chenopodium glaucum* (quadrats A1, A3, A5).

This community was mainly distributed in Langpiziliang Forest Plantations, Haojiaqiao Town, with an average importance value of 0.74 and an altitude of 1180 m. The average height of this community was 12.53–33.93 cm, and the total coverage was 30.3–49%. The dominant weeds were *Digitaria sanguinalis* and *Chenopodium glaucum*. The primary accompanying species included *Lappula heteracantha* and *Ephedra sinica*, with average importance values of 0.08.

Community XIII: *Digitaria sanguinalis + Chenopodium album* (quadrat A4).

This community was mainly distributed in Langpiziliang Forest Plantations, Haojiaqiao Town, with an average importance value of 0.76 and an altitude of 1180 m. The average height of this community was 14.22–30.98 cm, and the total coverage was 2.75–84.4%.

The dominant weeds were *Digitaria sanguinalis* and *Chenopodium album*. The primary accompanying species included *Ephedra sinica*, *Cynanchum chinense* and *Lactucatatarica*, with average importance values of 0.1, 0.08 and 0.06.

Community XIV: *Chenopodium glaucum + Digitaria sanguinalis + Ixeris chinensis* (quadrats A2, A19).

This community was mainly distributed in Langpiziliang Forest Plantations, Haojiaqiao Town, and Horticultural Farm of Dongta Town, with an average importance value of 0.78 and an altitude of 1170–1180 m. The average height of this community was 6.49–33.99 cm, and the total coverage was 5.9–41.5%. The dominant weeds were *Chenopodium glaucum*, *Digitaria sanguinalis* and *Ixeris chinensis*. The primary accompanying species included *Artemisia scoparia* and *Cynanchum chinense*, with average importance values of 0.22 and 0.1.

Community XV: *Chenopodium glaucum + Digitaria sanguinalis + Portulaca oleracea* (quadrats A6, A9, A10).

This community was mainly distributed in Langpiziliang Forest Plantations, Haojiaqiao Town, with an average importance value of 0.73 and an altitude of 1170 m. The average height of this community was 9.71–14 cm, and the total coverage was 4.5–71%.
The dominant weeds were *Chenopodium glaucum*, *Digitaria sanguinalis* and *Portulaca oleracea*. The primary accompanying species included *Lepidium apetalum*, *Echinochloa crusgalli* and *Sophora alopecuroides*, with average importance values of 0.11, 0.11 and 0.06.

## Relationship between the biodiversity of weed communities and soil properties

As shown in Fig. 3, the changing trends in the curves of Shannon-Wiener index (H), Simpson index (D) and the Pielou's index (E) were almost the same. Community X, with the highest H, and D of 1.64 and 0.79, respectively, was consisted of plot A36, and the dominant weeds were *Ixeris chinensis*, *Chenopodium glaucum*, and *Setariaviridis*. This community exhibited the characteristics of complex structure, more vegetation ecotypes and strong adaptability. The lowest H, D, E (1.14, 0.56, 0.71, respectively) was surveyed in community XIII, which consisted of plot A4, and the dominant weeds were *Digitaria sanguinalis* and *Chenopodium album*. Its height, low density and other unfavorable environmental factors would affect the growth of other plants.

The Spearman correlation coefficients between 15 weed community species diversity indices and 11 soil factors were analyzed (Fig. 4). Species diversity was more correlated to soil water content (SWC) and soil available potassium (AK) in comparison to other soil factors. AK had a significantly negative correlation with D ($p < 0.05$, r = −0.368), while SWC had a significantly positive correlation with E ($p < 0.05$, r = 0.372). H showed highly significant positive correlation with D, E and the species richness (S). Soil organic carbon (SOC) had a significantly positive correlation with total nitrogen (TN), total phosphorus (TP), total potassium (TK), available phosphorus (AP), AK and EC, whereas had a significantly negative correlation with pH. Meanwhile, SWC had a significantly positive correlation with SOC, TN, TP, TK, AP and EC.
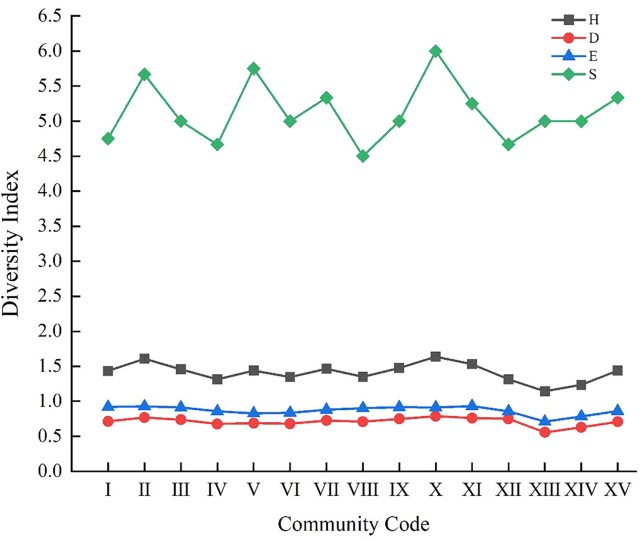

**Figure 3 Variation curve of species diversity of 15 communities.** H represents Shannon-Wiener index; D represents Simpson index; E represents Pielou's index; S represents species richness.

Multiple comparisons among all the environmental variables revealed that there were significant differences in environmental factors among the eight plots (Figs. 5A–5K). The plot S7 located in Lingwu Farm of Dongta Town, exhibited the highest contents of SOC, TN, TP, TK, AP, AK, EC, ammonium nitrogen ($NH_4^+$-N) and SWC in soil, but showed the lowest contents of pH and nitrate nitrogen ($NO_3^-$-N). The plot was largely dominated by the *Lepidium apetalum* community and *Ixeris chinensis* + *Amaranthus retroflexus* + *Chenopodium glaucum* communities. As can be seen, a substantial amount of water-soluble fertilizer was applied to soil. In contrast to plot S7, soil salinization was widespread in all other plots. Meanwhile, plots S1 and S4 had low levels of soil fertility and SWC. The plot S1 located in Langpiziliang Forest Plantations, Haojiaqiao Town, which was largely dominated by *Digitaria sanguinalis* + *Chenopodium glaucum* community. The plot S4 located in the Horticultural Farm of Dongta Town, which was largely dominated by *Chenopodium glaucum* + *Sophora alopecuroides* community. The plot S3, located in Daquan Forest Plantations, Haojiaqiao Town, exhibited the highest contents of $NO_3^-$-N, and the soil fertility also belonged to the upper-middle level. The amounts of $NO_3^-$-N and $NH_4^+$-N in soil can reflect the nitrogen supply of the soil (*Wang et al., 2019*). It is shown that plots S3 and S7 could be applied with little or no nitrogen. Proper soil fertility management is crucial to conservation agriculture.

## CCA ordinations

Based on the eleven soil factors, an environmental data matrix was established, and according to the importance values of species, a species data matrix was built. Then, preliminary in the DCA to test the models of species response to environmental variables,

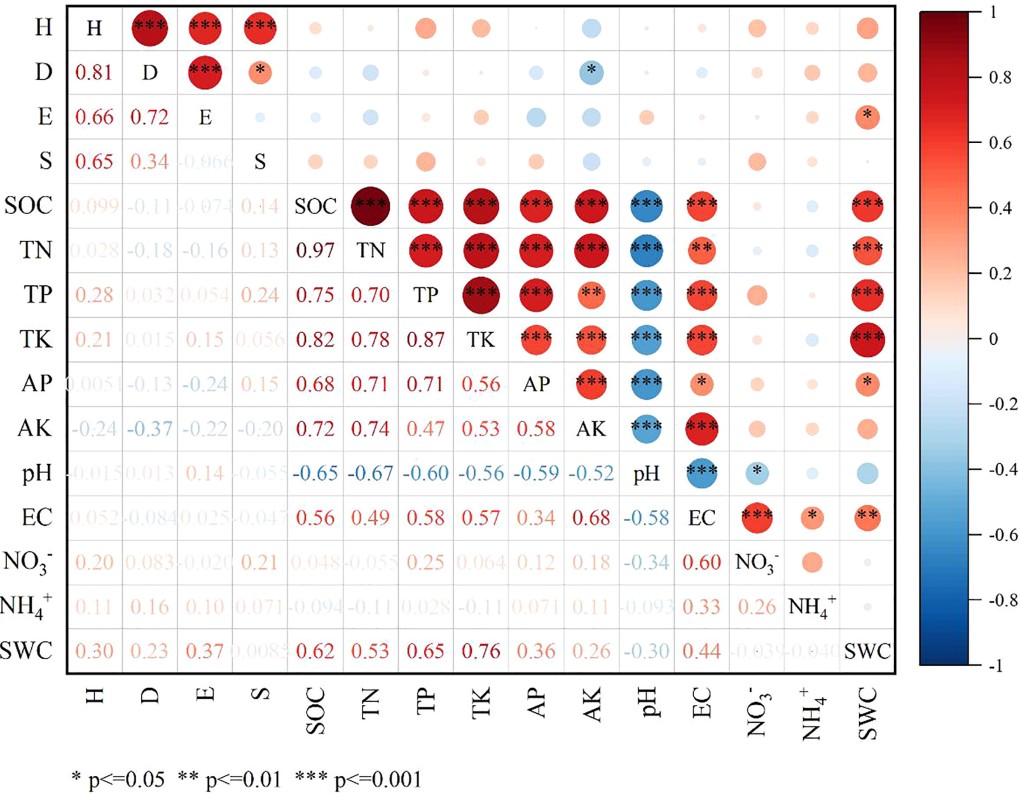

**Figure 4 Spearman correlation plot of diversity indexes and soil factors.**

the length of the gradient in the first axis was 4.4, demonstrating that the use of CCA as a unimodal model was appropriate. CCA was performed to evaluate the relationships between environmental factors and weed communities (Table 2 and Fig. 6). Our results showed that the first and second axes of the CCA ordination explained 41.47% of cumulative percentage variance in the species-environment relationship, with the eigenvalues of 0.4387 and 0.4147 respectively, and the species-environment correlations of 0.9276 and 0.9011 respectively. Moreover, the Monte Carlo test revealed that all canonical axes were significant (F = 1.5, $P$ = 0.002), which showed species distribution in the space was better visualized by the 11 selected environmental factors.

From Table 2, the first axis of CCA showed positive correlation with $NO_3^-$-N ($p < 0.05$), meanwhile showed extremely significant negative correlation with SWC ($p < 0.001$), the correlation coefficients were 0.311 and −0.5212 respectively. The second axis of CCA showed positive correlation with EC ($p < 0.05$), significantly positive correlation with SOC, TN, TP ($p < 0.01$) and extremely significant positive correlation with SWC ($p < 0.001$), the correlation coefficients were 0.3359, 0.3884, 0.4058, 0.3652 and 0.5376, respectively. Overall, SOC, TN, AP, AK, EC and SWC were the important environmental factors that affect the spatial distribution of species, followed by TP, TK and $NO_3^-$-N.

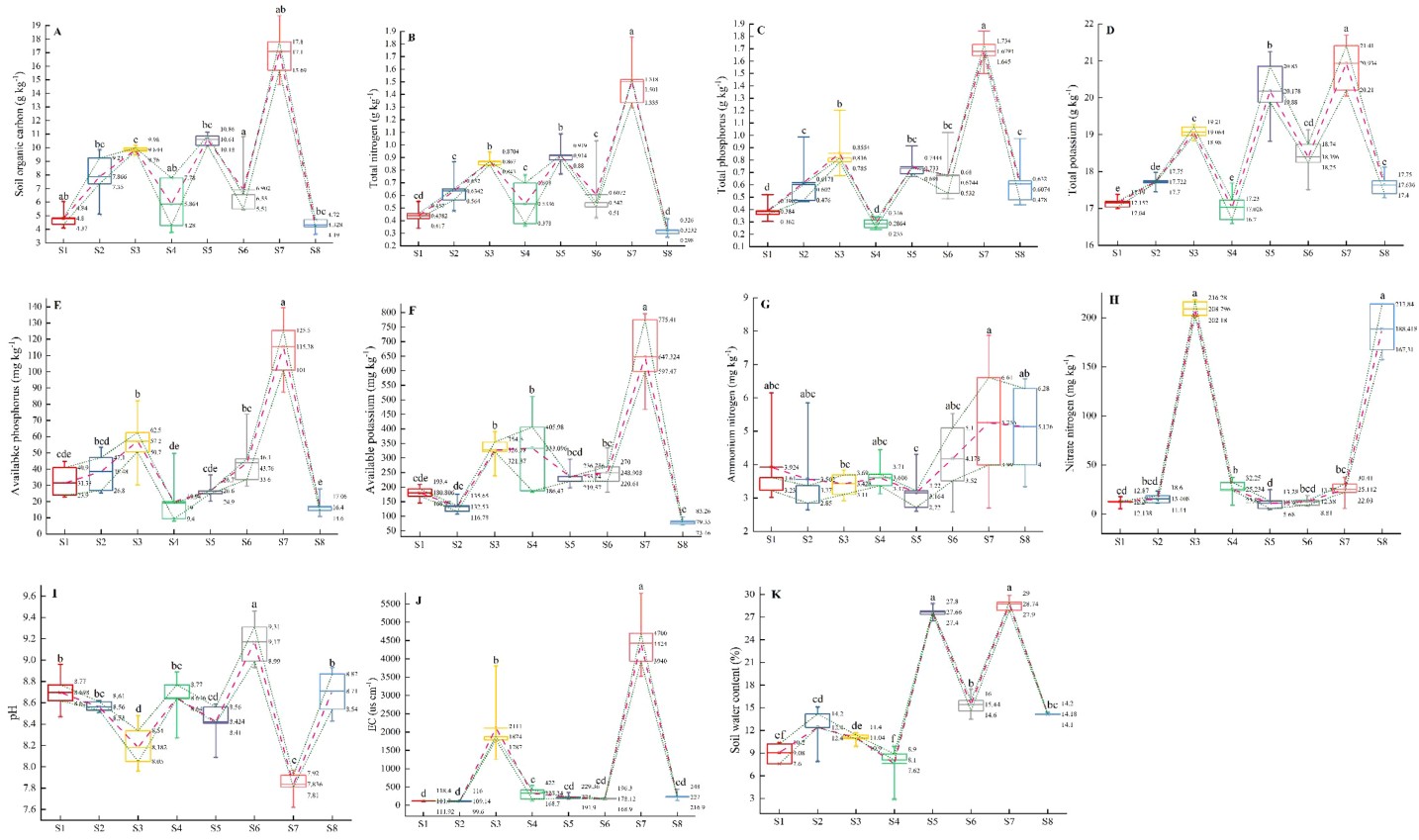

**Figure 5 (A–K) Multiple comparisons of environmental factors among weed communities in eight plots of 'LingwuChangzao' jujube.** Different lowercase letters (a–f) indicate statistical differences analyzed by one-way ANOVA and Duncan post-hoc test ($p < 0.05$). Box plots demonstrate medians (central lines), interquartile ranges (boxes) and ranges (whiskers). The connecting lines represent the mean and the 25th and 75th percentile.

The distribution of 40 quadrats in CCA was basically consistent with the results of TWINSPAN classification (Fig. 6), which reflected the environmental gradient of different weed communities. Community III (*Calystegia hederacea Wall*), community XII (*Ixeris chinensis + Amaranthus retroflexus + Chenopodium glaucum*) and community IX (*Ixeris chinensis + Agropyron cristatum*) were mainly distributed on the left of the first axis, suggesting that species tend to grew in a region of lower altitude, higher humidity and higher soil nutrients. Community V (*Rorippa islandica + Chenopodium glaucum*), Community VIII (*Chenopodium glaucum + Sophora alopecuroides*) and Community I (*Lactucatatarica*) were mainly distributed on the right of the first axis, that the habitat was vice versa of left. The second axis mainly reflected the change of SOC, TK, TN, TP and SWC. Community IV (*Lepidium apetalum*) was mainly distributed on the upper part of the second axis, suggesting that species tend to grow in a region of higher SOC, TN, TK, TP and SWC. Community XII (*Digitaria sanguinalis + Chenopodium glaucum*) and Community XV (*Chenopodium glaucum + Digitaria sanguinalis + Portulaca oleracea*) were mainly distributed on the lower part of the second axis, that the habitat was vice versa of the upper part of the axis.

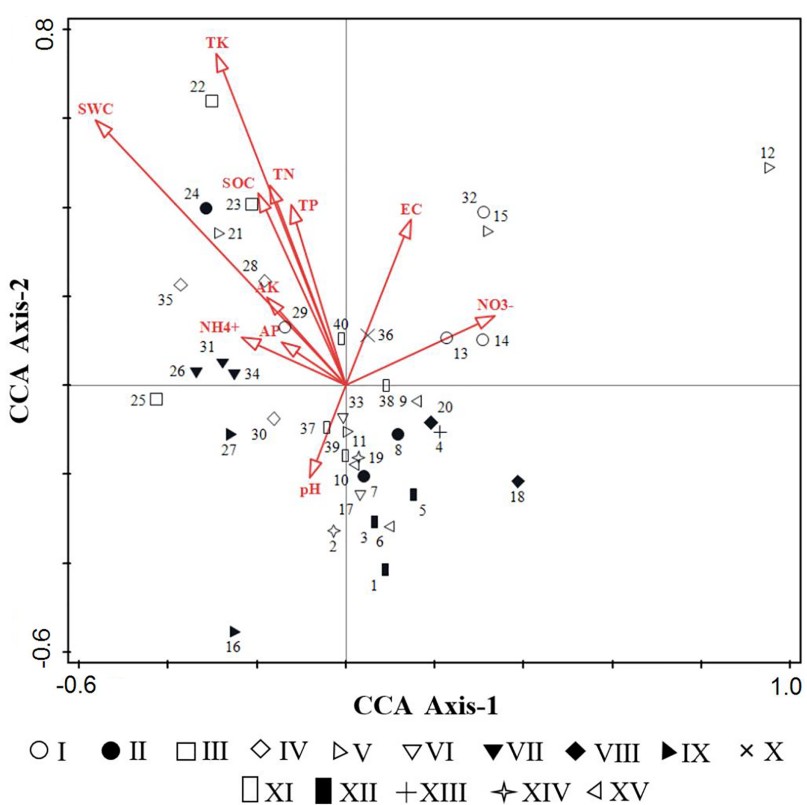

**Figure 6  CCA sequencing diagrams of weeds community quadrats and environmental factors.**

## DISCUSSION

### The relationship between species diversity and soil factors

Species diversity was often used as indicator for evaluating the dynamics, composition and structure of weeds (*Bai, Sadia & Yu, 2021*). The greater the diversity of weed species, the stronger the buffering effect on environmental changes or from population fluctuations within the weed community, which means that weeds are less prone to serious occurrence (*Niu & Niu, 2017*). The current study revealed that community X contains weed flora more than other communities that reflect this community exhibited the characteristics of complex structure, more vegetation ecotypes and strong adaptability. Some weed species exhibited broad ecological amplitude, while others were restricted to specific types of environments with very narrow amplitude, suggesting that the recorded ecological amplitude of weed species differed from each other caused both by phenotypic plasticity and heterogeneity (*Alsherif, 2020*). *Portulaca oleracea* is recognized as one of the worst weeds throughout the world and invades most subtropical and temperate agroecosystems, which occurs in almost all soil types and invades arable land quickly and is difficult to eradicate (*Visalakshy, 2007*). Chenopodium album is one of the problematic weeds, which has the characteristics of fast growth, high fecundity, wide adaptability and high phytotoxicity (*Alshallash, 2018*). Lactucatatarica and Calystegia hederacea Wall have been recorded as difficult weeds to control in subtropical orchards and rubber plantations, and

**Table 2 Correlation coefficient between the first four axes of CCA ranking and environmental factors.**

| Environment factor | Axis 1 | Axis 2 | Axis 3 | Axis 4 |
|---|---|---|---|---|
| Soil organic carbon, SOC | −0.1826 | 0.3884** | 0.2755 | −0.5474*** |
| Total nitrogen, TN | −0.1572 | 0.4058** | 0.2644 | −0.5562*** |
| Total phosphorus, TP | −0.1133 | 0.3652** | 0.3189* | −0.318* |
| Total potassium, TK | −0.2701 | 0.6718 | 0.1275 | −0.3404* |
| Available phosphorus, AP | −0.1334 | 0.088 | 0.3702** | −0.4381*** |
| Available potassium, AK | −0.1632 | 0.1771 | 0.535*** | −0.3341* |
| pH | −0.0746 | −0.1871 | −0.2696 | 0.6608 |
| EC | 0.1367 | 0.3359* | 0.4271*** | −0.2516 |
| Nitrate nitrogen, $NO_3^-$ | 0.311* | 0.1408 | 0.5897 | −0.0391 |
| Ammonium nitrogen, $NH_4^+$ | −0.2165 | 0.0968 | 0.1834 | 0.0221 |
| Soil water content, SWC | −0.5212*** | 0.5376*** | 0.0023 | −0.2452 |
| Summary of CCA ordination | | | | |
| Eigenvalue | 0.4387 | 0.4147 | 0.3599 | 0.2247 |
| Species-environment correlations | 0.9276 | 0.9011 | 0.8848 | 0.869 |
| Cumulative percentage variance of species-environment relation (%) | 21.56 | 41.94 | 59.63 | 70.67 |
| Significance test of all canonical axes | $F = 1.5$ $P = 0.002$ | | | |

Notes:
* $p < 0.05$.
** $p < 0.01$.
*** $p < 0.001$.

grow in many different crops in over 50 countries worldwide (*Misako, Keiko & Masahiro, 2005*). Consequently, the current research emphasizes the weed species, community diversity, environmental interpretation and damage to orchards, as well as the need to find solutions in the study area to reduce or eliminate them before harm occurs.

The composition of weed communities in orchards was influenced by factors, such as soil, climate, cultivation pattern, fertilization regimes, irrigation modalities and application of herbicides. Species distributions are sensitive to microhabitat changes on a small scale. The dominance of some weeds in the community may be modified by different factors. A large number of studies have shown that elevation, temperature, precipitations and soil factors are the dominant factors affecting vegetation diversity (*Karami-Kordalivand et al., 2021*; *Salama et al., 2017*), which is basically similar to our results. In addition, many weeds fit the integrated population model. They consist of many more or less spatially discrete populations. Each population constitutes a potential source of reproduction that may generate new populations (*Grice, 1998*), which will compete with jujube trees for soil nutrients and water, increase the temperature in the jujube orchard, and aggravate the damage of pests and diseases, cause yield reduction, and diminished quality (*Alsherif, 2020*). Consequently, applying niche theory to study the competition between weeds, selecting plant resources with strong niches and no adverse effects on fruit tree growth to occupy preferentially the niche cannot only effectively inhibit the growth of other weeds, but also have a longer-lasting effect. Meanwhile, an increasing number of studies show that the mineral nutrients (such as Ca, N, K, Fe, Mn, B) of the orchard soil play crucial

roles in fruit growth and quality (*Sun et al., 2022*). *Zhang et al. (2019)* reported that the firmness, pectin content, and nutrient substances of the calcium-treated jujube were maintained effectively since $Ca^{2+}$ is cross-linked with the carboxyl group in pectin, which can inhibit the decomposition of cell wall. As a result, we should strengthen the surveillance and control of the dominant species and the primary accompanying species of weed communities according to the breeding and growth characteristics of weeds to control the number of weeds effectively in the future. Simultaneously, the interspecific relationship between weed dominant species and 'LingwuChangzao' jujube was studied in-depth to evaluate the effect of different weed communities on the growth of jujube trees objectively, and thus providing a scientific basis for improving the yield and quality of fruit.

## The relationship between weed community distribution and soil factors

The growth and development of plants depend heavily on soil nutrient conditions, and the differences in the soil environment may cause changes in the community species diversity. *Goberna, Navarro-Cano & Verdu (2016)* found positive effects of soil fertility on plant richness, but the results of *Xue, Huang & Yu (2021)* showed an opposite trend that reducing soil fertility can promote the diversity of plant communities. Our results suggested that the diversity of weed communities in the main planting base of 'LingwuChangzao' jujube was positively correlated with EC and nitrate nitrogen. This might be because EC is an index for determining clay content, salt content, and mineralogy in soil, which is more suitable for plant growth and development (*Hao et al., 2021*). Nitrate nitrogen is a major N source available in aerobic soils and plays a critical role in root growth, which helps to alleviate the competition of weeds for nitrogen in the same soil layer (*Ma et al., 2021a*). The spatial distribution of plant communities is closely related to their environment, which is a comprehensive reflection of phenotypic plasticity and life history countermeasures under the influence of intra- and interspecific (*Lei et al., 2020*). The CCA ordination analysis can effectively perform statistical tests on multiple environmental indicators under different environmental gradients and thus better reflects the relation between the species diversity of the community and environment. In this study, CCA ordination showed that soil water content and soil total potassium were factors with the greatest effect on the weed communities, which is consistent with several previous studies (*Ibell, Xu & Blumfield, 2010*; *Neyret et al., 2018*). Moreover, the first two CCA axis explained the variance in the species-environment relationship, showing a comparatively better result. From Fig. 6, there is a significant correlation of soil water content with the first CCA axis, also reflecting corresponding changes in other soil factors and altitude. The second axis mainly reflected the change of total potassium, soil organic carbon, total phosphorus, and total nitrogen. It may be ascribed to the following points, (1) soil water acts as the most sensitive factor in plant ecosystems, not only affects plant growth and development but also determines vegetation types and limits vegetation distribution. (2) the variations over the altitudinal gradients indirectly induced heterogeneity of resources such as light, water and temperature, modifying the microclimate (*Bai et al., 2017*), which leads to changes in plant community structure.

(3) total potassium, soil organic carbon, total phosphorus, total nitrogen are the fundamental nutrients of the soil, and affect plant growth and development, *Zhang et al. (2012)* studied showed that there were strong correlations between plant communities and soil organic carbon, total phosphorus, total nitrogen and total potassium in Poyang Lake Nanji Wetland, which was consistent with our results.

Some weeds with allelopathic potential can release chemical substances (mainly secondary metabolites) into the environment through stem and leaf volatilization, leaching, root exudation and other pathways to promote or inhibit the growth and development of surrounding crops or other weeds. *Lactucatatarica* was distributed in areas with high available potassium, which may be due to the strong allelopathic effect of Compositae. A higher level of soil available potassium content helps synthesize a large amount of defensive allelopathic substances such as phenolics in plants (*Li et al., 2021*). *Digitaria sanguinalis* was distributed in the area with lower available nitrogen and available potassium in the CCA diagram, indicating that this plant has obvious avoidance behavior to high-level soil nutrients. Our study found that the underground stems of *Chenopodium glaucum* release secondary metabolites (allelochemicals) such as palmitic acid, sterol and phenol to inhibit the growth of surrounding plants, and thus most plants are far away in the two-dimensional CCA diagram (Fig. 6). However, the ecological distance among *Digitaria sanguinalis*, *Ixeris chinensis* and *Chenopodium glaucum* were close to each other on the CCA ordination diagram, indicating that they had similar spatial and nutritional niche, that is, they may have a strong antagonistic ability to deal with the allelopathic damage caused by *Chenopodium glaucum*. Previous studies showed that the extract of *Digitaria sanguinalis* can effectively inhibit the vegetative growth of *Chenopodium glaucum* (*Wang et al., 2017*). In conclusion, irrespective of which allelochemicals are produced and released by plants, most of them will eventually introduce into the soil where they can migrate and transform, and have an impact on soil fertility, the species, quantity and distribution of soil microorganisms, and further affect plant growth and fruit quality.

## CONCLUSIONS

The present study suggested that the weed communities in the main planting base of the 'LingwuChangzao' jujube can be divided into 15 association groups. There were significant differences in soil factors to the species diversity indices of the weed communities. The diversity of the weed communities increased with the decrease of available potassium, and the increase of soil water content. Results of TWINSPAN and CCA indicated that the community structure and spatial distribution of weed communities were affected by multiple environmental factors, such as soil water content, total potassium, which exhibited habitat preference. Our study contributed to understanding the alterations in diversity and community composition of weed along with the changes in environmental factors. Furthermore, our results provided a theoretical basis for weed invasion control and creating a higher biodiversity in arable land under the background of environmental change.

### Funding

This research was funded by the project of Key Research and Development Program of Ningxia Hui Autonomous Region (2019BFC02024), and the Major Science and Technology Special Project of Ningxia Hui Autonomous Region (2018BFH03015). There was no additional external funding received for this study. The funders had no role in study design, data collection and analysis, decision to publish, or preparation of the manuscript.

### Grant Disclosures

The following grant information was disclosed by the authors:
Project of Key Research and Development Program of Ningxia Hui Autonomous Region: 2019BFC02024.
Major Science and Technology Special Project of Ningxia Hui Autonomous Region: 2018BFH03015.

### Competing Interests

The authors declare that they have no competing interests.

### Author Contributions

- Xiaojia Wang performed the experiments, analyzed the data, prepared figures and/or tables, and approved the final draft.
- Bing Cao conceived and designed the experiments, authored or reviewed drafts of the article, and approved the final draft.
- Jin Zou conceived and designed the experiments, analyzed the data, prepared figures and/or tables, test help, and approved the final draft.
- Weijun Chen performed the experiments, authored or reviewed drafts of the article, provide space, and approved the final draft.

### Field Study Permissions

The following information was supplied relating to field study approvals (*i.e.*, approving body and any reference numbers):

Field experiments were approved by the Technical Service Center of Forestry and Fruit Tree of Lingwu City, Ningxia Province.

### Data Availability

The raw measurements are available in the Supplemental Files.

### Supplemental Information

Supplemental information for this article can be found online at http://dx.doi.org/10.7717/peerj.13583#supplemental-information.

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
