# Peer review of "Composition and environmental interpretation of the weed communities in the main planting base of jujube (Ziziphus jujuba Mill. cv. ‘LingwuChangzao’), Ningxia province of China"

_PeerJ, doi:10.7717/peerj.13583_

## Round 0.1 · original submission · Major Revisions

It is my opinion as the Academic Editor for your article - Composition and environmental interpretation of the weed communities in the main planting base of jujube (Ziziphus jujuba Mill. cv. ‘LingwuChangzao’), Ningxia province of China - that it requires a number of Major Revisions.

Reviewer 1 ·

Basic reporting

In general, the research content of this paper is novel. In this paper, the environmental factors and weed community diversity of the main planting bases of 'Lingwu Changzao' were analyzed by TWINSPAN and CCA. It is helpful to understand the relationship between weed distribution composition, distribution and environmental factors in jujube producing areas. The current version still needs some changes before Acceptance.
In the introduction, I noticed that the author mentioned the 'biodiversity protection' (Line 84-85), and I can't understand how it relates to this study. Is it to protect weeds?
Line 42, 'strategies. (He et al., 2020)' the period needs to be removed;
Line 55 and 374, 'Pan et al., (2015)' 'Zhang et al., (2012)' the comma needs to be removed;
Line 62, please quote the correct article, this article is not suitable for quoting here. I recommend you read this “Guo, M., Zhang, Z., Li, S., Lian, Q., Fu, P., He, Y., Qiao, J., Xu, K., Liu, L., Wu, M., Du, Z., Li, S., Wang, J., Shao, P., Yu, Q., Xu, G., Li, D., Wang, Y., Tian, S., Zhao, J., Feng, X., Li, R., Jiang, W. and Zhao, X. (2021) Genomic analyses of diverse wild and cultivated accessions provide insights into the evolutionary history of jujube. Plant Biotechnol J, https://doi.org/10.1111/pbi.13480”;
Line 65 to 108, Ma et al. 2021 has three articles in the bibliography, please mark them with a/b/c to distinguish them.
Line 71 72, lack of references after 'transcriptome differences analysis of fruit of grafting and root tiller propagation';
Line 96 and 103, missing space before parentheses;
Line 394 to 504, journal names in references should have a uniform format, some abbreviations and some full names.
Fig.3 is missing a note, 'H, D, E, S' should be noted;
Fig.7 to Fig.16 were not appear in the body of the article. Are they useful? Please mark it in the appropriate place!

Experimental design

no comment

Validity of the findings

no comment

Additional comments

no comment

Reviewer 2 ·

Basic reporting

The literature review and discussion should be improved.

Experimental design

Good

Validity of the findings

The current one is more on descriptive analysis. More in-depth reasoning should be applied to enhance the quality

Additional comments

Review for Peerj-71010-v0

The work contains some useful information, but the discussion was superficial which needs further in-depth elaboration and reasoning.

The analysis of the soil etc should finally be reflected in the fruit quality, which should be discussed further. For instance, whether the metal ions of the soil be finally linked to the contents in jujube fruit? For instance, Food Chemistry, 289, 40-48.

Many figures can be combined to make the data presentation concise and informative. The current ones are like presenting raw data. A key step is to extract the science behind rather than just simply presenting the data collected.

The composition and environmental interpretation should consider the approach of these influences which might be related to the metabolites, whether primary or secondary, but in total should be critical. Therefore, more metabolite investigations should be discussed further. For instance, Critical Reviews in Food Science and Nutrition, 61(9), 1448-1469.

More ecology reasons should be discussed on the effects of weed communities. For instance, Weed Science, 46, 467-474.

---

## Round 0.2 · accepted · Accept

Please carefully check each chart according to the format requirements of the journal and whether each table is referenced in the paper. In particular, check all the references are properly formatted.

Reviewer 2 ·

Basic reporting

The authors addressed the questions quite well.

Experimental design

Good

Validity of the findings

Good.

Additional comments

The authors have improved the manuscript. The current version is acceptable for publication.